# An Updated Analysis of Exon-Skipping Applicability for Duchenne Muscular Dystrophy Using the UMD-DMD Database

**DOI:** 10.3390/genes15111489

**Published:** 2024-11-20

**Authors:** Jamie Leckie, Abdullah Zia, Toshifumi Yokota

**Affiliations:** 1Department of Medical Genetics, Faculty of Medicine and Dentistry, University of Alberta, Edmonton, AB T6G 2H7, Canadaazia2@ualberta.ca (A.Z.); 2The Friends of Garrett Cumming Research & Muscular Dystrophy Canada HM Toupin Neurological Sciences Research, Edmonton, AB T6G 2H7, Canada

**Keywords:** Duchenne muscular dystrophy (DMD), exon-skipping, antisense oligonucleotide (ASO), dystrophin, Becker muscular dystrophy (BMD), applicability

## Abstract

Background/Objectives: Antisense oligonucleotide (ASO)-mediated exon-skipping is an effective approach to restore the disrupted reading frame of the dystrophin gene for the treatment of Duchenne muscular dystrophy (DMD). Currently, four FDA-approved ASOs can target three different exons, but these therapies are mutation-specific and only benefit a subset of patients. Understanding the broad applicability of exon-skipping approaches is essential for prioritizing the development of additional therapies with the greatest potential impact on the DMD population. This review offers an updated analysis of all theoretical exon-skipping strategies and their applicability across the patient population, with a specific focus on DMD-associated mutations documented in the UMD-DMD database. Unlike previous studies, this approach leverages the inclusion of phenotypic data for each mutation, providing a more comprehensive and clinically relevant perspective. Methods: The theoretical applicability of all single and double exon-skipping strategies, along with multi exon-skipping strategies targeting exons 3–9 and 45–55, was evaluated for all DMD mutations reported in the UMD-DMD database. Results: Single and double exon-skipping approaches were applicable for 92.8% of large deletions, 93.7% of small lesions, 72.4% of duplications, and 90.3% of all mutations analyzed. Exon 51 was the most relevant target and was applicable for 10.6% of all mutations and 17.2% of large deletions. Additionally, two multi-exon-skipping approaches, targeting exons 45–55 and 3–9, were relevant for 70.6% of large deletions and 19.2% of small lesions. Conclusions: Current FDA-approved ASOs were applicable to 27% of the UMD-DMD population analyzed, leaving a significant portion of patients without access to exon-skipping therapies. The clinical translation of alternative approaches is critical to expanding the accessibility of these therapies for the DMD population.

## 1. Introduction

Duchenne Muscular Dystrophy (DMD) is a severe and progressive genetic disease resulting in the degeneration and necrosis of muscle tissue that has a significant effect on patient life expectancy [1]. The DMD phenotype is the result of genetic mutations, typically frameshift mutations, in the dystrophin gene that prevent the production of functional dystrophin [2,3]. Although life expectancy has improved with advances in medical management and a better understanding of the disease [4], there is a critical need for effective therapies that can significantly improve patient quality of life. Antisense oligonucleotides (ASOs) are short, single-stranded nucleotides that are capable of binding to a specific region of RNA to promote exon-skipping [5]. ASO-mediated exon-skipping is a particularly attractive approach to treating DMD as it allows for the exclusion of a specific exon or exons to restore a disrupted reading frame [6,7].

Since frameshift mutations associated with the DMD phenotype can occur throughout the gene, each exon-skipping strategy is mutation(s)-specific and is only applicable to a subset of DMD patients [8,9,10]. Currently, there are four FDA-approved ASOs for the treatment of DMD that target exon 45, 51, or 53, which address the most common DMD mutations [11,12]. However, many patients are left without access to FDA-approved exon-skipping therapies, highlighting the need for the further development of ASOs targeting other exons.

Ongoing studies are exploring exon-skipping therapies for some of these additional exons to broaden the reach of these treatments [13,14,15]. Understanding which exons could be targeted to benefit the largest number of patients can help researchers prioritize development efforts. In 2009, Aartsma-Rus et al. provided an overview of all single and double exon-skipping approaches for DMD based on mutations reported in the Leiden DMD mutation database [16]. Bladen et al., in 2015, reported the top 10 exon-skipping targets with the highest applicability, using data from the TREAT-NMD DMD Global database [17]. More recently, Lim et al. (2020) assessed the applicability of these top 10 approaches, along with two multi-exon-skipping approaches targeting mutation hotspots, by analyzing 414 DMD-associated mutations in the Canadian population [11]. A key advantage of this study is its use of the UMD-DMD database, which includes over 2800 reported DMD mutations and provides the associated phenotype for each [18]. This is an improvement over previous studies that relied on the Leiden and TREAT-NMD DMD Global database, which lack phenotypic data. As a result, some *DMD* gene mutations included in the previous analysis may not be appropriate targets for exon-skipping strategies due to their association with a milder phenotype. The inclusion of non-DMD-associated mutations could affect the accuracy of the overall applicability previously reported for these strategies. Furthermore, as diagnostic techniques improve and genetic testing becomes easier to access, our understanding of the specific mutations that result in DMD may have changed over time [19,20]. This review provides an updated overview of all ASO-mediated single and double exon-skipping strategies, as well as two multi-exon-skipping approaches, and their potential applicability, specifically in the DMD patient population, as reported in the UMD-DMD database [18].

## 2. Background on Duchenne Muscular Dystrophy

DMD is a fatal genetic disease that affects between 1 in 5000 and 1 in 6000 live male births [21,22]. Symptom onset typically begins only two to three years after birth, with early symptoms involving muscle weakness and difficulty walking [23]. Most patients with DMD develop wheelchair dependency by the age of 12, and symptoms of cardiomyopathy present in the patients around 18 years of age [24]. DMD has a serious effect on patient life expectancy, typically due to respiratory and cardiac failure [4,25]. While advances in medical management have led to a significant increase in the lifespan of DMD patients over the last few decades, the average life expectancy for patients born after 1990 is still only 28.1 years [4,26].

DMD is an X-linked recessive genetic disease that is caused by mutations in the dystrophin (*DMD*) gene, the largest gene in the human genome, spanning roughly 2.3 megabases and 79 exons [27,28]. The full-length dystrophin protein is 427 kDa, although several shorter isoforms have been identified [29]. Dystrophin is composed of an actin-binding amino-terminal domain (ABD1), a central rod domain, a cysteine-rich domain, and a carboxyl terminus [30]. Within skeletal muscles, dystrophin localizes to the sarcolemma, where it plays an important role in the dystrophin-glycoprotein complex (DGC) [31]. As part of the DGC, dystrophin anchors the actin cytoskeleton to the extracellular matrix, providing the sarcolemma with protection against the forces involved in muscle contractions [32,33].

DMD is most often associated with mutations in the *DMD* gene that disrupt the protein’s reading frame and introduce a premature stop codon [34]. As a result of these mutations, the DMD transcripts and the truncated dystrophin protein are subject to decay and degradation, respectively. Consequently, patients with DMD have little to no functional dystrophin [23]. In the absence of dystrophin, skeletal, diaphragm, and cardiac muscle becomes particularly prone to contraction-induced damage, and their cell membrane become leaky [29,33]. Additionally, the loss of dystrophin impairs cell signalling and the organization of other DGC-associated proteins [35]. This leads to a cycle of chronic muscle degeneration and regeneration in muscle cells undergoing frequent contractions [1]. Due to impaired repair processes that are characteristic of DMD, the deposition of excess extracellular matrix components, known as fibrosis, occurs at these sites of injury [29,36]. The progressive fibrotic deposition that occurs leads to impaired muscle function and progressive weakness [37]. The standard of care for all DMD patients currently includes long-term corticosteroid use to slow disease progression by mitigating inflammation-induced muscle damage and promoting muscle regeneration [38,39].

In contrast, mutations in the *DMD* gene that do not disrupt the reading frame are typically associated with a milder phenotype known as Becker muscular dystrophy (BMD) [8,34]. Patients with BMD possess partially functional, truncated dystrophin proteins that are associated with later disease onset and slower disease progression compared to DMD [40]. There is substantial phenotypic variability between patients with BMD. The onset of symptoms can occur anywhere between childhood and adulthood [41,42]. The symptoms that present can vary from limb-girdle myopathy and quadricep myopathy to hyperCKemia accompanied by normal strength [43]. While most BMD patients have asymptomatic cardiac involvement, some patients may develop dilated cardiomyopathy and subsequent heart failure [44]. The prognosis for BMD is much better than for DMD, and patient life expectancy is between 40 and 50 years [42], and in some cases the disease has little impact on life expectancy [45].

## 3. ASOs for Exon-Skipping

ASOs are versatile molecules composed of 15 to 25 nucleotides that are capable of sequence-specific binding to target RNA, forming an RNA-ASO hybrid [46]. Once bound, ASOs can regulate gene expression or promote alternative splicing to treat disease [47]. Unmodified nucleotides are highly susceptible to nuclease degradation, and their size and associated charge can make it difficult for them to penetrate cell membranes [48,49]. To overcome these challenges, several chemical modifications have been identified that improve the stability and cellular uptake of ASOs, typically involving altering the internucleotide linkage and/or the nucleotide’s sugar moiety [50]. Depending on the desired mechanism of the ASO, these modifications can be applied to some or all the nucleotides within the ASO.

ASOs can alleviate the effect of disease-causing mutations through various mechanisms usually categorized into RNase H1-dependent and RNase H1-independent (RNA blockage) [51]. RNase H1 is an enzyme that plays an important role in maintaining DNA stability by cleaving RNA when it forms a hybrid with DNA during replication or repair [52]. ASOs aimed at reducing gene expression can utilize the endogenous role of RNase H1 by binding to target RNA to induce cleavage [53]. To be recognized by RNase H1, these ASOs, known as gapmers, are composed of a central region of unmodified nucleotides, flanked by chemically modified nucleotides [54]. Alternatively, ASOs can be fully modified to induce steric hindrance or modulate splicing [55]. These ASOs can be used to block the interaction between target RNA and ribosomal subunits to prevent protein translation [56]. They can also target pre-mRNA splice junctions, exonic splicing enhancers (ESEs), or exonic splice silencers (ESSs) to block splicing factors and induce alternative splicing [5]. Exon-skipping, mediated by ASOs, is a promising approach for the treatment of many rare genetic diseases. The reading frame of out-of-frame proteins can be restored using ASOs that are designed to induce the exclusion of specific exons possessing and/or surrounding a frameshift deletion to restore the reading frame [57,58]. ASOs can also promote the exclusion of exons containing pathogenic gain-of-function mutations [59]. The primary goal of exon-skipping is to facilitate the production of a truncated protein that is at least partially functional to improve patient symptoms.

When designing ASOs for exon-skipping, it is important to note that exons act as puzzle pieces that fit together in specific ways to create proteins [60]. The three nucleotides that make up each codon in the open reading frame sometimes exist on different exons if they are separated by an intron [61]. Introns can be categorized into three different phases based on if or where they interrupt the codon. If the surrounding introns are in the same phase, the exon is symmetrical [62]. Alternatively, if the phases are different for the surrounding introns, the exon is considered asymmetrical. If a large deletion occurs in a single or series of symmetrical exons, the reading frame should remain intact as the remaining exons surrounding the deletion will still fit together and maintain the open reading frame. However, if a large deletion involves an asymmetrical exon, or multiple exons that start and end at differing phases, the reading frame will become disrupted, which has the potential to be restored through exon-skipping (Figure 1) [63]. Small lesions that exist in a symmetrical exon can be excluded by promoting the exclusion of the mutated exon. However, if the mutation occurs in an asymmetrical exon, skipping an adjacent exon, or exons, is required to maintain the reading frame.

### ASO-Mediated Exon-Skipping for the Treatment of DMD

Exon-skipping has proven to be an effective approach for increasing levels of partially functional dystrophin in DMD patients. The frameshift mutations that commonly cause DMD are ideal targets for ASOs, which can promote the skipping of specific exons around the deletion to restore the disrupted reading frame [64]. This approach would result in a truncated DMD protein. However, there is a large range of multi-exon in-frame deletions that are associated with reduced levels of truncated dystrophin, resulting in the significantly milder BMD phenotype [42]. These findings indicate that many exon-skipping approaches have the potential to convert the more severe DMD phenotype into the milder BMD phenotype by partially restoring dystrophin levels [64,65].

There are currently several FDA-approved ASOs available for the treatment of DMD, targeting three different exons to restore the reading frame of frame-shifted transcripts. The first FDA-approved ASO for the treatment of DMD, Eteplirsen, which induces the skipping of exon 51, received FDA approval in 2016 [66]. The FDA approval of Golodirsen and Viltolarsen, which both target exon 53, and Casimersen, targeting exon 45, followed in the years after [12]. All currently approved ASOs for DMD are phosphorodiamidate morpholino oligomers (PMOs), composed of morpholine rings connected through phosphorodiamidate linkages [67]. The chemical modifications inherent in PMOs protect them from nuclease degradation [68]. They are neutrally charged, highly stable, and associated with minimal toxicity, even after multiple doses [69,70]. While PMOs have demonstrated only suboptimal efficacy in non-dystrophic muscles, they show significantly greater cellular uptake in dystrophic skeletal muscles [71], likely due to the heightened regenerative activity in these tissues [72]. Overall, PMOs represent a safe and efficient approach to promoting exon-skipping in DMD patients.

All four FDA-approved treatments were observed to effectively promote the skipping of their target exon and increase the levels of partially functional dystrophin in treated patients [73,74,75,76]. Functional benefits have also been observed in clinical trials for Eteplirsen and Golodirsen, where disease progression in treated patients appeared to be attenuated in comparison to controls, as assessed through a 6-min walk test [75,77]. However, controversy persists regarding the overall effectiveness of these therapies, given the small size of treatment groups and the observation that dystrophin protein levels remain below what is believed to be necessary for significant clinical improvement and enhanced survival [66,78,79]. Currently, multiple clinical trials are underway to further assess the long-term effectiveness and safety of these therapeutics [80].

The current protocol for administering these ASOs to DMD patients involves weekly intravenous injections [74,81]. Given that DMD impacts skeletal muscles throughout the body, systemic delivery via intravenous injection is beneficial. However, ASOs must traverse the bloodstream and cross multiple biological barriers to reach their target tissues [82]. Furthermore, PMOs are subject to rapid renal clearance, which complicates their distribution [83]. Despite high-dose and frequent injections, ASO uptake remains inconsistent across different muscle groups and is particularly limited in cardiac tissue [84,85].

To enhance the efficacy of ASOs for DMD, alternative chemical modifications are being explored in clinical trials, with the goal of reducing the required dose and frequency of injections. One promising candidate is WVE-N531, which targets exon 53 and is composed of a phosphoryl guanidine-containing (PN) backbone, offering improved pharmacokinetics and the unique capability to cross the blood–brain barrier [86,87]. Alternatively, DS-5141B, which promotes exon 45 skipping, uses 2’-O,4’-C-ethylene-brdiged nucleic acid (ENA) modifications that demonstrated superior uptake in both skeletal and cardiac muscle when compared to PMOs [88,89]. Despite showing no safety concerns in phase 1/2 clinical trials, the development of DS-5141B was recently discontinued [90]. Continued advancements in ASO chemical modifications have the potential to greatly enhance their therapeutic effectiveness for DMD patients.

## 4. Applicability of Exon-Skipping for Duchenne Muscular Dystrophy

This review aims to provide an updated overview of the theoretical applicability of all exon-skipping approaches for the treatment of DMD, and their applicability in the DMD population. Although there are several FDA-approved ASOs for the treatment of disease, they are applicable exclusively for exon 45, exon 51, and exon 53 skipping [12]. These targets are only applicable to a portion of the DMD population because these therapies are mutation-specific. Numerous additional targets of exon-skipping have been identified to be potentially disease-modifying due to their ability to restore disruptions in the dystrophin reading frame [16]. Understanding the applicability of current and future approaches can guide the direction of future research.

### 4.1. Genetic Mutations Associated with Duchenne Muscular Dystrophy

Thousands of mutations throughout the *DMD* gene have been identified as being associated with the DMD phenotype [10]. While various mutation types have been observed, large deletions in the DMD gene are the most common cause, accounting for an estimated 60–70% of cases [8,9,91]. The remaining DMD-causing mutations consist of duplications (5–15%) and small mutations (20%), including point mutations, small insertions, and small deletions [17,66]. Although these mutations have been identified throughout the *DMD* gene, there are specific regions that are recognized as mutational hotspots. Exons 43–55 are estimated to be the site of over 70% of known deletions, and exons 2–22 contain over 20% of the observed deletions [92,93].

This review will focus on the large deletions, large duplications, and small lesions recorded in the UMD-DMD database (www.umd.be/DMD/), accessed on 8 October 2024, a French database that collected molecular and clinical data from patients with DMD gene mutations [18]. A total of 1901 large deletions were identified within the database, with each case classified into specific phenotypic groups. Of these records, 1028 were associated with DMD, while the remaining 873 were associated with BMD, intermediate muscular dystrophy (IMD), dilated cardiomyopathy (DCM), or cases pending diagnosis. Among the 1028 large deletions, 61 (5.9%) of the records involved in-frame deletions, while 967 (94%) involved frameshift deletions. Additionally, the UMD-DMD database reports 633 small lesions, which include small deletions and insertions (<1 exon), splice site mutations, and point mutations, as well as 334 large duplications. Of the duplications, 232 were associated with DMD, including 52 single-exon duplications and 180 multi-exon duplications. For small lesions, 457 were associated with DMD. Overall, in this patient population, 60% of DMD-related mutations are large deletions, 27% are small lesions, and 13.5% are large duplications.

### 4.2. Applicability of Exon-Skipping for Large Deletions

All 967 large deletions associated with DMD that result in a frameshift were assessed for the potential applicability of exon-skipping to restore the *DMD* gene’s reading frame. Figure 2 visually represents the 79 exons of the dystrophin protein and their corresponding intron phases, serving as a tool to predict how exons might align before and after applying the exon-skipping methods discussed [29]. The 61 in-frame deletions were excluded from evaluation since they do not possess a disruption in their reading frame that exon-skipping can restore. Each large deletion was analyzed for the presence of exons immediately before and/or after the deletion that, if skipped, could restore the reading frame. The analysis of all mutation types (large deletions, small lesions, and duplications) considered the potential for single or double exon-skipping approaches, as well as multi-exon-skipping of exon 3–9 and exon 45–55.

All exon-skipping strategies identified as applicable to frameshift large deletions, as well as small lesions and duplications, associated with DMD in the UMD-DMD database are summarized in Table 1 (single or double exon-skipping) and Table 2 (multi exon-skipping). A total of 58 exon-skipping strategies were found to be relevant for the reported large deletions, including the two multi (>2)-exon-skipping strategies. Of these, 31 involved skipping a single exon, while 25 required skipping two exons to restore the dystrophin protein’s reading frame. The most applicable single or double exon-skipping strategies were exon 51 (17.2%), exon 45 (15.1%), exon 53 (13.7%), and exon 44 (10.7%). For multi-exon-skipping, the approaches targeting exon 45–55 and exon 3–9 were applicable to 60.9% and 9.7% of large deletions, respectively (Table 2), with exon 45–55 skipping emerging as the most broadly applicable approach for large deletions.

The total number of records in Table 1 exceeds the total reported DMD mutations in the UMD-DMD database because many of these mutations can be corrected using multiple different exon-skipping strategies. For example, the frameshift caused by the c.7543_7660del mutation, which leads to the deletion of exon 52, can be corrected by skipping exon 51, exon 53, or exons 45–55. Once a mutation-specific exon-skipping therapy receives FDA approval, other potential exon targets for the same mutation become less relevant for further study, as patients would already have access to treatment and be expected to achieve a similar therapeutic outcome.

The overall applicability of exon-skipping strategies for the large deletions associated with DMD, in addition to small lesions and duplications, was assessed and reported in Table 3. Of the reported large deletions, 81.7% could have their reading frame restored by promoting the exclusion of a single exon. Double exon-skipping and multi-exon-skipping strategies were applicable for 13.3% and 70.6% of the reported deletions, respectively. Altogether, 92.8% of the reported large deletions can theoretically have their reading frame restored through either single, double, or multi-exon-skipping. For frameshift deletions specifically, the exon-skipping strategies were applicable to 98.3% of cases.

### 4.3. Applicability of Exon-Skipping for Small Lesions

The exclusion of an intra-exonic small lesion can typically be achieved by skipping the affected exon if it is symmetrical, or by skipping both the affected exon and an adjacent exon if the affected exon is asymmetrical. Small lesions located within splice sites generally result in the deletion of a neighboring exon [16]. Mutations in potential donor sites, near the 3’ end of an exon, and acceptor sites, near the 5’ end of an exon, were analyzed as if they resulted in exon skipping of either the downstream or upstream exon, respectively. A total of 66 single and double exon-skipping approaches were identified as applicable to patients with small lesions in the UMD-DMD database (Table 1). Of the reported small lesions, 47.9% could be excluded through single exon-skipping and 45.7% could be excluded through double exon-skipping, for a total of 93.7% of the small lesions amenable to these approaches (Table 3). The most applicable single and double exon-skipping targets for small lesions were exons 69 and 70 (6.6%), exons 58 and 59 (3.9%), and exons 19 and 20 (3.7%). Multi-exon-skipping approaches are applicable to 19.2% of the reported patients (Table 2). Additionally, there is minimal overlap between therapies suited for small lesions versus those for large deletions or duplications. Therefore, distinct ASOs will need to be developed and tested to provide exon-skipping therapies for the majority of this subgroup.

### 4.4. Applicability of Exon-Skipping for Large Duplications

Exon-skipping is theoretically highly effective for frameshift single exon duplications by targeting the duplicated exon for exclusion from the final transcript. Although ASOs cannot typically distinguish between the original and duplicated exon, skipping either exon should correct the effects of the duplication [94,95]. A total of 14 single exon-skipping approaches were determined to be applicable for the 52 single exon duplications reported (Table 1). Exon 2 has previously been identified as the most frequently duplicated exon in DMD [96], and skipping exon 2 was found to be the most applicable approach for duplications, with the potential to correct 16.8% of duplications. While previous studies have shown that ASOs can effectively skip one copy of the duplicated exon [94,95,97], some ASOs have been observed to promote the exclusion of both copies, which can lead to a disrupted reading frame [94]. This disruption can often be corrected by skipping an adjacent exon. Notably, 78.8% of the single exon duplications have an adjacent exon that can be skipped to restore the reading frame if both the original and duplicated exons are excluded.

Exon-skipping strategies to amend multi-exon-skipping duplication remain a much more complicated approach than single-exon duplications [94]. In theory, targeting one or two exons belonging to the duplication can restore the reading frame. Of the 180 multi-exon duplications associated with DMD in the UMD-DMD database, 113 (62.8%) could theoretically be restored through single or double exon-skipping of the duplicated exons. However, if the original exon(s) is mistargeted with the ASOs and is excluded, the reading frame will be disrupted, making these ASOs a technically challenging approach that requires further proof of concept [94].

### 4.5. Overall Applicability and Clinical Application of ASO-Mediated Exon-Skipping for DMD

Overall, 115 exon-skipping strategies were identified as applicable to the UMD-DMD patient population analyzed. Figure 3 illustrates the applicability of the top 10 single and double exon-skipping strategies, as well as the overall applicability of all identified approaches. Exon-skipping was most relevant for large deletions, followed by small lesions. Single and double exon-skipping approaches were found to be applicable for 90.3% of all reported mutations associated with DMD (Table 3), surpassing the 83% applicability reported by Aartsma-Rus et al. (2009) based on the Leiden DMD database [16]. This increased applicability is likely due to the exclusion of non-DMD-associated mutations in the current analysis. Specifically, single exon-skipping was potentially applicable to 69.2% of mutations, while double exon-skipping could be applied to an additional 22.9% of mutations. The multi-exon-skipping strategies, targeting exons 3–9 and exons 45–55, were deemed applicable to 47.4% of patients, which is lower compared to the 60% applicability reported in the Canadian DMD population by Lim et al. (2020) [11]. The higher potential applicability in the Canadian cohort may be partially attributed to a greater proportion of patients with large deletions (69%), the mutation type for which these multi-exon-skipping were determined to be most relevant. Additionally, the applicability of specific approaches has been shown to vary between studies analyzing different DMD databases. For example, exon 53 skipping ranks fourth in the Canadian population but second in the TREAT-NMD and Leiden DMD populations [11,16,17]. In the UMD-DMD population, exon 53 skipping was ranked third, following exon 45 skipping.

The three most widely applicable single exon-skipping strategies identified, targeting exon 45, 51, and 53, are the targets of the currently FDA-approved ASO therapies for DMD. These exons have consistently been identified as the top three targets in previous studies, with the exception of exon 53 skipping in the Canadian DMD population [11,16,17]. These therapies could potentially benefit 27% of the total patient population; 43.4% of patients reported large deletions, 5.2% reported duplications, and 1.1% of the patients reported small lesions. This leaves a substantial portion of the patient population who have mutations that could theoretically be addressed by exon-skipping, without access to approved ASO therapy. Further research evaluating the effectiveness of these therapies for restoring partial dystrophin function is expected to increase their accessibility for DMD patients.

The applicability of exon-skipping in this review is theoretical and based solely on the potential restoration of the reading frame caused by frameshift mutations. The reading frame rule, which suggests that DMD and BMD are the result of out-of-frame and in-frame mutations, respectively, in the dystrophin gene [34] is not always correct. Although they may represent only a minority of both populations, BMD patients can possess out-of-frame mutations and DMD patients can possess in-frame mutations [3]. As such, some of the deletions resulting from exon-skipping may not lead to an improved phenotype, regardless of the restoration of the reading frame. Understanding the phenotype associated with the total deletion that would result from exon-skipping enables developers to predict how effective their therapy will be in converting the DMD phenotype to the milder BMD phenotype [2]. Furthermore, exon-skipping approaches may lead to a truncated transcript that has not been reported in the patient population, which may not achieve the required functionality to improve symptoms, due to the exclusion of indispensable regions or impaired protein folding [30].

For all newly developed ASOs, evaluating their ability to improve symptoms associated with the DMD phenotype in DMD cell and/or mouse models is essential for clinical translation [98]. Since these therapies are mutation-specific, DMD mouse models have been created to replicate mutations relevant to these treatments. For example, the mdx52 mouse model, which carries a deletion in exon 52, has been used to assess the effectiveness of ASO-mediated exon 51 and exon 53 skipping strategies [99,100]. For in vitro analysis, the efficacy of ASO therapies for DMD is typically evaluated using patient-derived myoblasts and human-induced pluripotent stem cell-derived cardiomyocytes, as these cells represent the primary therapeutic targets [101,102].

Alternative emerging therapeutic strategies, now accessible to patients through clinical trials or recent FDA approvals, primarily focus on the delivery of micro-dystrophin via adeno-associated virus (AAV) vectors to promote the production of truncated, yet functional, dystrophin. One such gene therapy, Elevidys, has recently received FDA approval [103], with ongoing clinical trials exploring the use of alternative micro-dystrophin constructs and different AAV serotypes [104]. These therapies have demonstrated effectiveness in increasing dystrophin-positive muscle fibers, leading to improved ambulation in patients with just a single injection [104,105]. Since some of these approaches are not mutation-specific, they could potentially benefit a broad portion of the DMD patient population [29]. However, the effects of ASO-mediated exon-skipping following AAV-mediated micro-dystrophin treatment remain unclear. Given the potential overlapping applicability of these therapies, it is crucial to confirm the absence of potential adverse effects before administering ASOs to patients who have previously received micro-dystrophin therapy.

The multi-exon-skipping approaches of exon 3–9 and 45–55 were included in the analysis as they target exons belonging to the mutation hotspots of the dystrophin gene [11]. Exon 45–55 skipping was determined to be the overall most applicable approach for large deletions and small lesions. The skipping of exon 3–9 was also the most applicable exon-skipping approach for large deletions targeting outside the exon 43–55 mutation hotspot. Each approach represents a single cocktail therapy with a wider range of applicability than alternative single or double exon-skipping strategies targeting the same region (Figure 4) [11,106]. Furthermore, deletions of exon 3–9 and 45–55 have been identified in patients with mild BMD or asymptomatic phenotypes [107,108,109], suggesting that these regions are not essential for function. These strategies are an appealing approach for those looking to develop widely applicable exon-skipping therapies for DMD. However, there are additional clinical hurdles to overcome for these cocktail ASO therapies as each ASO component, as well as the complete cocktail, must prove to be safe for FDA approval, adding significant costs to their development [110].

Although ASO-mediated exon-skipping has received clinical approval for the treatment of DMD, they remain limited in their ability to increase partially functional dystrophin levels in muscle and have little effect on cardiac tissues [111]. The suboptimal efficacy of these therapies in patients is largely attributed to challenges associated with poor uptake into cells and endosomal entrapment [112,113]. Currently, a large focus of those researching ASOs has been aimed at developing and testing strategies to improve cellular uptake. One such approach is using non-viral delivery systems, such as lipid nanoparticles and extracellular vesicles, that can encapsulate the ASO and effectively deliver its cargo across cell membranes [114,115,116]. An alternative approach is using bioconjugates that can interact with specific cell receptors to enhance delivery to target tissues and promote cellular uptake. There are many promising bioconjugates that have been identified, including cell-penetrating peptides (CPPs), cholesterol, aptamers, and antibodies [117]. CPPs, which can be conjugated to neutrally charged PMOs, have shown great promise in DMD mouse models. Specific peptides have been identified as targeting cardiac tissues, leading to a significant improvement in dystrophin expression in cardiac muscles [118,119,120]. Overall, the ongoing efforts to improve ASO delivery to target tissues are likely to lead to the identification of therapeutic strategies that will improve their efficacy at lower doses and mitigate the cardiac phenotype in DMD patients.

## 5. Conclusions

In this review, ASO-mediated exon-skipping was determined to have the potential to amend the pathogenic effect of most large deletions, small lesions, and single exon duplications associated with DMD. However, only four of these therapies have been approved, leaving many DMD patients without access to potentially disease-modifying therapies. The comprehensive list of all potential targets, and their associated applicability to the DMD population, have been provided with the goal of guiding future ASO development. Further research evaluating the functionality of the exon-skipping of different applicable exons, and strategies to improve their delivery, are required to not only improve accessibility in the DMD patient population but also to increase their efficacy at a safe dose for long-term clinical use.

## Figures and Tables

**Figure 1 genes-15-01489-f001:**
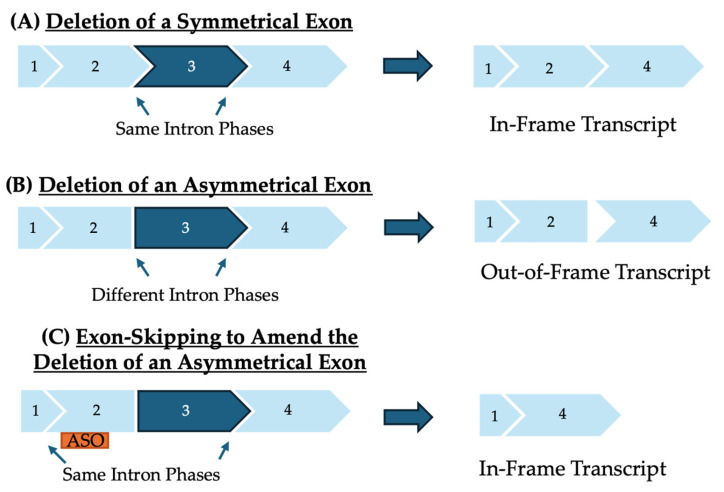
(**A**) Symmetrical exons are surrounded on either end by the same intron phase so, if they are deleted, the remaining transcript still fits together, and the reading frame is maintained. (**B**) Asymmetrical exons have a different intron phase on either end. When an asymmetrical exon is deleted, the remaining exons no longer fit together, disrupting the reading frame for all following exons. (**C**) Exon-skipping can restore the reading frame that has been disrupted by a deletion of an asymmetrical exon by promoting the exclusion of an addition exon, so the remaining exons can fit together.

**Figure 2 genes-15-01489-f002:**
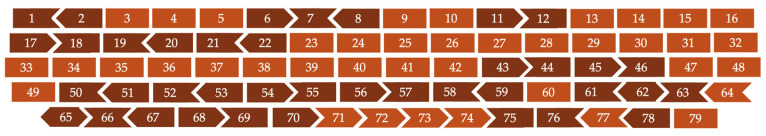
Visual representation of all 79 exons of the full-length dystrophin transcript and their respective intron phases (not to scale). Symmetrical exons are shown in orange and asymmetrical exons are shown in dark red.

**Figure 3 genes-15-01489-f003:**
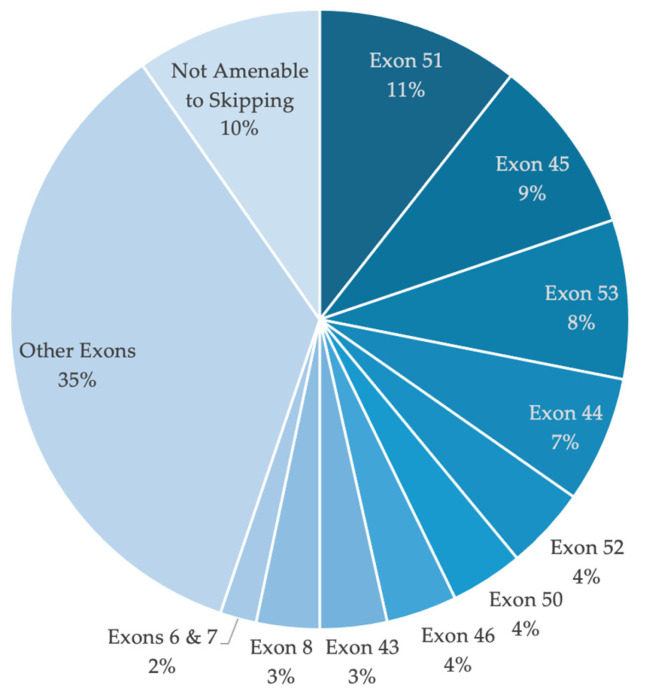
Pie chart depicting the applicability of all identified exon-skipping strategies for the treatment of DMD in the total DMD patient population. The chart highlights the top 10 single and double exon-skipping targets based on their applicability.

**Figure 4 genes-15-01489-f004:**
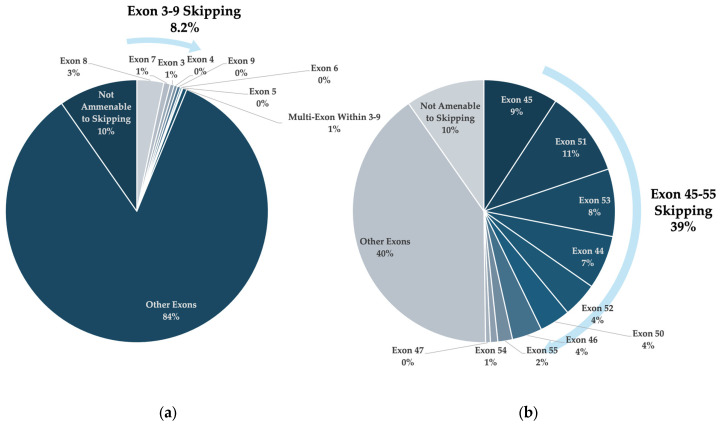
Pie charts illustrating the applicability of multi-exon-skipping approaches in the total DMD patient population, compared to single exon-skipping strategies targeting within the same region. (**a**) shows the overall applicability of exon 3–9 skipping, and (**b**) shows the applicability of exon 45–55 skipping, as indicated by the light blue arrows.

**Table 1 genes-15-01489-t001:** Overview of the applicability of all single or double exon-skipping approaches for all large deletions, small lesions, and duplications associated with DMD reported in the UMD-DMD database. Each exon’s applicability was determined in the population of all 1717 DMD records.

Ranking	Exon(s)	All Mutations	Deletions	Small Lesions	Duplications
1	51	10.6%	17.2%	0.7%	2.2%
2	45	9.1%	15.1%	0.4%	1.3%
3	53	8.3%	13.7%		1.7%
4	44	6.7%	10.7%	0.4%	3.0%
5	52	4.2%	6.5%		3.0%
6	50	3.8%	6.2%		1.3%
7	46	3.7%	5.7%		1.3%
8	43	3.3%	5.5%		0.4%
9	8	3.3%	4.7%		8.2%
10	6 & 7	2.9%	3.9%		10.3%
11	2	1.8%	2.9%		16.8%
12	55	1.8%	3.7%	0.2%	1.3%
13	12	1.7%	2.1%	1.3%	1.7%
14	69 & 70	1.6%		6.6%	
15	53 & 54	1.5%	1.8%		0.4%
16	45 & 51	1.3%	1.8%		
17	58 & 59	1.3%		3.9%	
18	19 & 20	1.3%		3.7%	
19	68 & 69	1.2%		3.7%	
20	11	1.2%	1.6%		0.4%
21	20 & 21	1.1%		3.5%	
22	17	1.1%	1.5%		0.4%
23	52 & 53	1.1%		3.3%	
24	21 & 22	1.0%		3.1%	
25	17 & 18	1.0%		2.8%	
26	35	1.0%		2.8%	
27	22	0.9%	1.2%		0.9%
28	21	0.9%	1.2%		0.4%
29	54	0.9%	1.2%		
30	11 & 12	0.9%		2.6%	
31	23	0.9%		2.4%	
32	51 & 52	0.8%		2.4%	
33	7	0.8%	1.0%		0.4%
34	34	0.8%		2.2%	
35	43 & 44	0.8%		2.2%	
36	20	0.8%	0.8%	1.3%	0.4%
37	56	0.7%	0.6%		3.4%
38	18	0.7%	0.6%		3.0%
39	54 & 55	0.7%		2.0%	
40	55 & 56	0.6%		2.0%	
41	75 & 76	0.6%		2.0%	
42	19	0.6%	0.8%		
43	52 & 55	0.6%	0.8%		
44	15	0.6%		1.8%	
45	40	0.6%		1.8%	
46	47	0.6%		1.8%	
47	7 & 8	0.6%	0.9%		0.9%
48	64	0.5%		1.8%	
49	46 & 47	0.5%	0.7%		
50	3	0.5%	0.2%	1.1%	
51	10	0.5%		1.5%	
52	14	0.5%		1.5%	
53	26	0.5%		1.5%	
54	44 & 45	0.5%		1.5%	
55	12 & 13	0.5%	0.6%		0.9%
56	18 & 19	0.5%	0.6%		
57	2 & 7	0.5%	0.6%		
58	16	0.5%		1.3%	
59	62	0.4%	0.3%	0.2%	0.4%
60	50 & 51	0.4%		1.3%	
61	59 & 60	0.4%	0.5%		
62	4	0.4%		1.1%	
63	9	0.4%		1.1%	
64	37	0.4%		1.1%	
65	39	0.3%		1.1%	
66	45 & 46	0.3%		1.1%	
67	56 & 57	0.3%		1.1%	
68	65 & 66	0.3%		1.1%	
69	69	0.3%		1.1%	
70	21 & 44	0.3%	0.4%		
71	50 & 55	0.3%	0.4%		
72	65	0.3%	0.1%	0.9%	
73	25	0.3%		0.9%	
74	28	0.3%		0.9%	
75	30	0.3%		0.9%	
76	48	0.3%		0.9%	
77	62 & 63	0.3%		0.9%	
78	69	0.2%		0.9%	
79	59	0.2%	0.3%	0.2%	
80	24	0.2%		0.7%	
81	27	0.2%		0.7%	
82	33	0.2%		0.7%	
83	41	0.2%		0.7%	
84	42	0.2%		0.7%	
85	74	0.2%		0.7%	
86	6	0.2%	0.1%		1.7%
87	10 & 11	0.2%			2.2%
88	63	0.2%	0.2%		
89	8 & 9	0.1%	0.2%		
90	57 & 62	0.1%	0.2%		
91	5	0.1%		0.4%	
92	13	0.1%		0.4%	
93	36	0.1%		0.4%	
94	38	0.1%		0.4%	
95	60	0.1%		0.4%	
96	49 & 50	0.1%			1.7%
97	61	0.1%	0.1%		
98	2 & 3	0.1%	0.1%		
99	42 & 43	0.1%	0.1%		
100	17 & 20	0.05%	0.1%		
101	17 & 22	0.05%	0.1%		
102	61 & 62	0.05%	0.1%		
103	50 & 57	0.05%	0.1%		
104	57 & 58	0.05%	0.1%		
105	64 & 65	0.05%	0.1%		
106	63 & 64	0.05%	0.1%		
107	66	0.05%	0.1%		
108	68	0.05%	0.1%		
109	66 & 67	0.05%	0.1%		
110	51 & 63	0.05%	0.1%		
111	29	0.05%		0.2%	
112	31	0.05%		0.2%	
113	32	0.05%		0.2%	
114	58	0.05%		0.2%	
115	60 & 61	0.05%			0.9%

**Table 2 genes-15-01489-t002:** Overview of the applicability of multi-exon-skipping approach for large deletions and small lesions associated with DMD mutations in the UMD-DMD database.

Ranking	Exons	All Mutations	Deletions	Small Lesions
1	45–55	39.2%	60.9%	10.3%
2	3–9	8.2%	9.7%	8.9%

**Table 3 genes-15-01489-t003:** Summary of the applicability of single and double exon-skipping for large deletions, small lesions, and duplications associated with DMD in the UMD-DMD database. The total applicability was determined using all DMD-associated mutations in the database.

	Single Skipping	Double Skipping	Single and/or Double Skipping	Multi (>2) Skipping
Deletions	81.7%	13.3%	92.8%	70.6%
Small Lesions	47.9%	45.7%	93.7%	19.2%
Duplications	56.0%	20.7%	72.4%	-
All Mutations	69.2%	22.9%	90.3%	47.4%

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
