# Peer review of "An Updated Analysis of Exon-Skipping Applicability for Duchenne Muscular Dystrophy Using the UMD-DMD Database"

_genes, 2024, doi:10.3390/genes15111489_

Round 1
Reviewer 1 Report
Comments and Suggestions for Authors
In this manuscript, the authors have provided a comprehensive review on the status of ASO in the treatment of DMD. The manuscript is well written and the authors have discussed the ASO in details. This review is important for the field. I have some minor comments which are provided below
1. The authors didnot provide the details about the use of the AAV in treating DMD patients in the various clinical trials.
2. The authors should provide the details about the stability of various ASO used in clinical trials.
Author Response
Response to Reviewer One:
In this manuscript, the authors have provided a comprehensive review on the status of ASO in the
treatment of DMD. The manuscript is well written and the authors have discussed the ASO in
details. This review is important for the field. I have some minor comments which are provided
below
Comment 1: The authors did not provide the details about the use of the AAV in treating DMD
patients in the various clinical trials.
Response: Thank you for pointing this out. We have included a section on the use of AAV in
treating DMD on line 408-421 that reads: “Alternative emerging therapeutic strategies, now
accessible to patients through clinical trials or recent FDA approvals, primarily focus on delivery
mico-dystrophin via adeno-associated virus (AAV) vectors to promote the production of
truncated, yet functional, dystrophin. One such gene therapy, Elevidys, has recently received
FDA approval, with ongoing clinical trials exploring the use of alternative micro-dystrophin
constructs and different AAV serotypes. These therapies have demonstrated effectiveness in
increasing dystrophin-positive muscle fibers, leading to improved ambulation in patients with
just a single injection. Since some of these approaches are not mutation-specific, they could
potentially benefit a broad portion of the DMD patient population. However, the effects of ASO-
mediated exon-skipping following AAV-mediated micro-dystrophin treatment remain unclear.
Given the potential overlapping applicability of these therapies, it is crucial to confirm the
absence of potential adverse effects before administering ASOs to patients who have previously
received micro-dystrophin therapy.” Relevant references have been included.
Comment 2: The authors should provide the details about the stability of various ASO used in
clinical trials.
Response: We appreciate your valuable suggestion. A section on the stability of various ASOs in
clinical trials has been included on line 223-232, which reads: “To enhance the efficacy of ASOs
for DMD, alternative chemical modifications are being explored in clinical trials, with the goal
of reducing the required dose and frequency of injections. One promising candidate is WVE-
N531, which targets exon 43 and is composed of a phosphoryl guanidine-containing (PN)
backbone, offering improved pharmacokinetics and the unique capability to cross the blood-brain
barrier. Alternatively, DS-5141B, which promotes exon 45 skipping, uses 2’-O,4’-C-ethylene-
brdiged nucleic acid (ENA) modifications that demonstrated superior uptake in both skeletal and
cardiac muscle when compared to PMOs. Despite showing no safety concerns in phase 1/2
clinical trials, the development of DS5141B was recently discontinued. Continued advancements
in ASO chemical modifications have the potential to greatly enhance their therapeutic
effectiveness for DMD patients.” References have been included.
Response to Reviewer Two:
I read with interest the manuscript of Leckie and collegues and I found it very well written and
interesting.
The authors provide an overview on Exon skipping strategies fo Duchenne Muscular distrophy
pointing out pros and cons of this strategy.
I congratulate with the authors for the clarity of their work which is relevant and worth
publishing.
I have some comments that could help improve the mansucript.
Comment 1: In the discussion section, correctly the authors point out the limits of this approach.
In particular the authors underline the delivery issues. I would add a section about delivery wich
is not described in the text.
Response: Thank you for the valuable suggestion to include a section about delivery. We have
included a section on ASO delivery for the treatment of DMD on lines 216-222, which reads:
“The current protocol for administering these ASOs to DMD patients involves weekly
intravenous injections. Given that DMD impacts skeletal muscles throughout the body, systemic
delivery via intravenous injection is beneficial. However, ASOs must traverse the bloodstream
and cross multiple biological barriers to reach their target tissues. Furthermore, PMOs are subject
to rapid renal clearance, which complicates their distribution. Despite high-dose and frequent
injections, ASO uptake remains inconsistent across different muscle groups and is particularly
limited in cardiac tissue.” References have been included.
Reviewer 2 Report
Comments and Suggestions for Authors
I read with interest the manuscript of Leckie and collegues and I found it very well written and interesting.
The authors provide an overview on Exon skipping strategies fo Duchenne Muscular distrophy pointing out pros and cons of this strategy.
I congratulate with the authors for the clarity of their work which is relevant and worth publishing.
I have some comments that could help improve the mansucript.
1. In the discussion section, correctly the authors point out the limits of this approach. In particular the authors underline the delivery issues. I would add a section about delivery wich is not described in the text.
2. Often in the text, it is referred to the necessity to test new ASOs in a cellular or mouse model. I would add a brief description of which cellular models are used to this end and a brief description of which mouse models are currently used for DMD and BMD.
3. I would add a sentence or two about the concurrent therapies that patients receive other than ASO.
4. I would rewrite the sentence at line 65-68, to be more clear.
5. In Figure 1 consider to center the image and better spacing from the text. Why not include the case of same phase exons just for completeness?
6. Line 273 check for correct spacing.
Author Response
Response to Reviewer Two:
I read with interest the manuscript of Leckie and collegues and I found it very well written and
interesting.
The authors provide an overview on Exon skipping strategies fo Duchenne Muscular distrophy
pointing out pros and cons of this strategy.
I congratulate with the authors for the clarity of their work which is relevant and worth
publishing.
I have some comments that could help improve the mansucript.
Comment 1: In the discussion section, correctly the authors point out the limits of this approach.
In particular the authors underline the delivery issues. I would add a section about delivery wich
is not described in the text.
Response: Thank you for the valuable suggestion to include a section about delivery. We have
included a section on ASO delivery for the treatment of DMD on lines 216-222, which reads:
“The current protocol for administering these ASOs to DMD patients involves weekly
intravenous injections. Given that DMD impacts skeletal muscles throughout the body, systemic
delivery via intravenous injection is beneficial. However, ASOs must traverse the bloodstream
and cross multiple biological barriers to reach their target tissues. Furthermore, PMOs are subject
to rapid renal clearance, which complicates their distribution. Despite high-dose and frequent
injections, ASO uptake remains inconsistent across different muscle groups and is particularly
limited in cardiac tissue.” References have been included.
Comment 2: Often in the text, it is referred to the necessity to test new ASOs in a cellular or
mouse model. I would add a brief description of which cellular models are used to this end and a
brief description of which mouse models are currently used for DMD and BMD.
Response: We appreciate the suggestion to include a brief description on the cellular models are
used to evaluate the ASO therapies discussed. We have included an explanation of the cellular
and mouse models used on lines 401-407, which reads: “Since these therapies are mutation
specific, DMD mouse models have been created to replicate mutations relevant to these
treatments. For example, the mdx52 mouse model, which carries a deletion in exon 52, has been
used to assess the effectiveness of ASO-mediated exon 51 and exon 53 skipping strategies. For
in vitro analysis, the efficacy of ASO therapies for DMD is typically evaluated using patient-
derived myoblasts and human induced pluripotent stem cell-derived cardiomyocytes, as these
cells represent the primary therapeutic targets.” References for the use of these models in the
evaluation of treatment effectiveness have been included.
Comment 3: I would add a sentence or two about the concurrent therapies that patients receive
other than ASO.
Response: Thank you for the valuable suggestion to include information about concurrent
therapies that patients receive other than ASOs. On line 113-115, a description of the current
standard of care was included that reads: “The standard of care for all DMD patients currently
includes long-term corticosteroid use to slow disease progression by mitigating inflammation-
induced muscle damage and promoting muscle regeneration.” In addition, the description of
emerging gene therapies was included on line 408-421 that reads: “Alternative emerging
therapeutic strategies, now accessible to patients through clinical trials or recent FDA approvals,
primarily focus on delivery mico-dystrophin via adeno-associated virus (AAV) vectors to
promote the production of truncated, yet functional, dystrophin. One such gene therapy,
Elevidys, has recently received FDA approval, with ongoing clinical trials exploring the use of
alternative micro-dystrophin constructs and different AAV serotypes. These therapies have
demonstrated effectiveness in increasing dystrophin-positive muscle fibers, leading to improved
ambulation in patients with just a single injection. Since some of these approaches are not
mutation-specific, they could potentially benefit a broad portion of the DMD patient population.
However, the effects of ASO-mediated exon-skipping following AAV-mediated micro-
dystrophin treatment remain unclear. Given the potential overlapping applicability of these
therapies, it is crucial to confirm the absence of potential adverse effects before administering
ASOs to patients who have previously received micro-dystrophin therapy.” Relevant references
have been included.
Comment 4: I would rewrite the sentence at line 65-68, to be more clear.
Response: We appreciate your valuable suggestion. The sentence in line 68-71 has been update
from “Furthermore, as diagnostic techniques for accurately identifying DMD-causing mutations
continue to improve and genetic testing becomes more accessible, the mutation landscape for
DMD may have evolved” to “Furthermore, as diagnostic techniques improve and genetic testing
becomes easier to access, our understanding of the specific mutations that result in DMD may
have changed over time”
Comment 5: In Figure 1 consider to center the image and better spacing from the text. Why not
include the case of same phase exons just for completeness?
Response: Thank you for pointing this out. Figure one has been centered and has been provided
with additional spacing from the text. The case of a deletion occurring in a symmetrical exon is
shown in Figure 1. a. As these deletions lead to an in-frame transcript, they are not applicable for
exon-skipping.
Comment 6: Line 273 check for correct spacing.
Response: We appreciate your comment on the correct spacing of line 273. This line has been
confirmed to have the correct spacing.